# A FAIR Semantic Data Marketplace for the Smart City of Tomorrow

André Pomp, Andreas Burgdorf, Alexander Paulus, and Tobias Meisen

Chair of Technologies and Management of Digital Transformation,
University of Wuppertal, Wuppertal, Germany
{pomp, burgdorf, paulus, meisen}@uni-wuppertal.de

**Abstract.** Today, smart city applications are largely based on data collected from different stakeholders. This paper presents a semantic data marketplace that aims at publishing and consuming data in a findable, accessible, interoperable and reusable (FAIR) way. It is based on the principles of semantic data management, i.e., data providers annotate their added data with semantic models.

## 1 Introduction

Smart cities are becoming more and more of a reality every day. While the first autonomous vehicles are making their way into cities on a trial basis, parking spaces can already report whether they are occupied or free, sensors monitor the air quality, and multi-modal apps also allow people to get around using a variety of different mobility solutions. What all these applications have in common is that they are based on data that is generated and managed at different locations by different stakeholders. However, in order to derive real added value from this data, for example by combining it in order to develop new applications, this data must be findable, accessible, interoperable and reusable (FAIR). Although the FAIR principles [4] have essentially become a central element of research data management, their use for the data management within smart cities is quite limited. While there is an increasing push for open data to be available as Linked Open Data (LOD)[1], publishing data in the LOD does not imply that it follows the FAIR principles [3]. Especially when enterprise data or data from citizens should be published, the use of FAIR principles becomes more important. In order to enable a better data basis for the smart city, a system is needed that allows the publishing and consuming of data from different stakeholders under the consideration of the FAIR principles.

For addressing this issue, we propose a semantic data marketplace that we develop in the ongoing research project *Bergisch.Smart_Mobility*[2]. This marketplace allows cities, enterprises, and citizens to share their data with all interested stakeholders. In this way, we want to ensure that data which may be of interest for smart city applications follows FAIR principles across city boundaries

---

[1] https://www.w3.org/DesignIssues/LinkedData.html
[2] https://www.bergischsmartmobility.de/en/the-project/

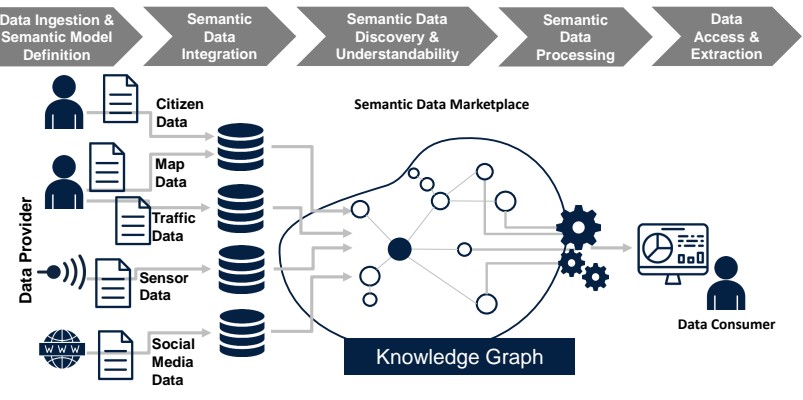

**Fig. 1.** Schematic overview of the workflow for providing and consuming data on the semantic data marketplace.

so that applications based on existing data in one city will work in other cities as well. Inspired by projects such as KARMA [2] or Optique [1], the developed data marketplace is based on the principles of semantic data management, where provided data is annotated with semantic models based on a conceptualization. In addition to the common challenges of designing a software architecture that is scalable and extensible and can be used for a wide variety of heterogeneous batch and streaming data in a wide variety of data formats, the following challenges in the area of semantic data management were particularly important:

(i) The creation of an initial conceptualization that can be used for creating semantic models.
(ii) The definition of a process that allows the continuous (automatic) evolution of the common shared conceptualization as not all concepts and relations are available in the beginning or are part of different ontologies.
(iii) The implementation of algorithms that support the (semi-)automatic creation of semantic models.
(iv) The design of a user-friendly way to enable domain experts, which do not have any experience with semantic technologies, to create and refine the semantic models for their data.
(v) The definition of a process that allows data consumers to extract the data from the marketplace exactly in the way that they need it for their use case.

In the following, we present the semantic data marketplace that tackles these challenges.

## 2   The Semantic Data Marketplace

In general, the semantic data marketplace distinguishes between the role of a data provider and that of a data consumer (cf. Figure 1). The data provision

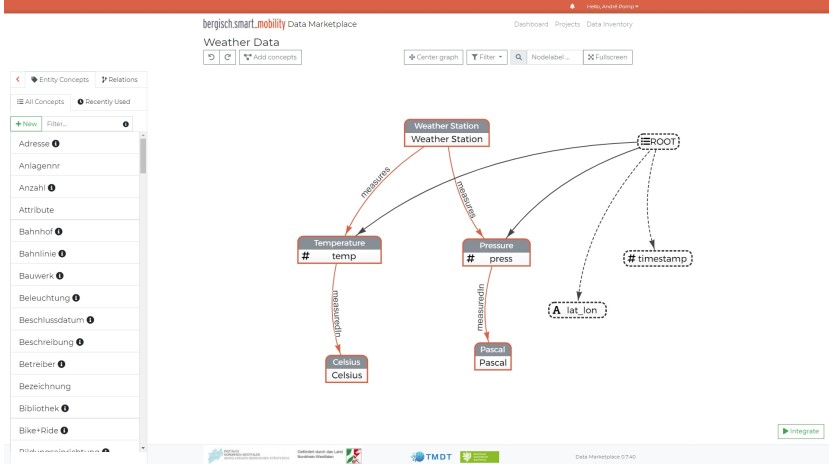

**Fig. 2.** Overview of the user interface that allows domain experts to refine semantic models using a drag and drop interaction.

process consists of providing the corresponding data, e.g., by uploading a file or defining a source at which data can be accessed, just as a database, an FTP server or an MQTT stream. Based on the data provided, the semantic data marketplace analyzes the structure, i.e., the schema of the data, and automatically creates an initial semantic model. The generated semantic model is then displayed to the data provider in the user interface (cf. Figure 2). Here, the data provider is now given the opportunity to refine the model. Possible user operations in this phase are adding new concepts or relations from the conceptualization as well as deleting or replacing existing elements that may have been wrongly identified by the algorithm for the automatic semantic model creation. Compared to traditional Ontology-based Data Management, where the conceptualization is static, the semantic data marketplace allows the data providers to introduce own concepts and relationships if necessary. This enables the continuous evolution of the internal conceptualization. As soon as the data provider has finished the creation of the semantic model, the data is converted into an internal common data format and integrated into a central data lake. In addition, the semantic model is stored in a knowledge graph, which combines the used conceptualization as well as all semantic models created so far in one single structure. The data itself only resides within the data lake.

Based on the knowledge graph and the associated semantic models, data consumers can find and understand data by searching for semantic concepts. The retrieved data can then be accessed and retrieved by data consumers in one of three data formats (CSV, XML, JSON). Batch data sets can be downloaded whereas streaming data can be subscribed to in order to receive them in real-time. The semantic data marketplace was already used within a hackathon[3],

---

[3] https://rethinking-mobility.de/hack4sc/technik/

in which teams had to develop smart city applications based on 77 data sets provided by three city departments from three different cities.

## 3    Demonstration

The demonstration will show the data provision and consumption workflow including the semantic model creation based on a typical Bergisch.Smart_Mobility use case covering a geospatial batch data set and a streaming data source. Afterwards, an additional data set is provided that should be integrated into the platform by the audience. The demonstration will be performed based on the live instance of the semantic data marketplace[4]. An example video for a typical smart city scenario can be found here[5].

## 4    Conclusion

In this paper, we presented our semantic data marketplace that allows cities, enterprises, and citizens to share their data with all interested stakeholders considering the FAIR principles. While the focus in the first half on the project was mainly on enabling the data provision by domain experts, the remainder of the project will focus on the data consumption including the definition of semantic processing pipelines and data format extractions based on template engines. In this way it will become possible to consume data exactly the way it is required by data consumers for their use case.

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
