# OpenReview forum: "A FAIR Semantic Data Marketplace for the Smart City of Tomorrow"
_eswc-conferences.org/ESWC/2021/Conference/Poster_and_Demo_Track — Submitted to ESWC2021 P&D_

### Official Review · AnonReviewer2 · 2021-04-08
**Many unclear things and not a clear advance in the state of the art**

**Rating:** 4
**Confidence:** 3

**Review:**

The paper proposes a semantic data hub in a form of a marketplace which is envisaged to solve the information silos problem in the smart cities of tomorrow. It provides data publishers and consumers with a nice user interface where to operate and fulfil their data necessities. Although I find the idea quite promising I have some major concerns:

You allow data providers to upload their data and allow them to refine the semantic model, and then you say that in contrast to a traditional Ontology-based Data Management data providers can change the underlying schema dynamically. So if users can build their own models how you perform then the integration of datasets? My concern here is that the integration of data is not literally mentioned or shown in the demo. So, I cannot assume that you are integrating data.

Then, if data is not integrated you are just making a simple translation from one format to another, which by the way is what we can see in the demo video. Therefore, my concern is what added value can deliver your platform in comparison to the mentioned Karma or other approaches like RMLEditor. One could think that is just a layer on top of these existing systems.

You also show in Fig. 1 that you use a Knowledge Graph as the back-end of your application. But it is not exposed to users and moreover you only offer non-semantic formats like CSV, XML and JSON as an output. I think that because of it you are neglecting Interoperability and Reusability of the mentioned FAIR principles. Additionally, not exposing your KG aren't you promoting an information silo?

In the linked video I would suggest to slow it down a little bit. This was something that I had to do to understand what was happening in the screen. Also, it would really benefit from an oral presentation.

Out of curiosity, you say that the platform analyses the data structure. How it performs that? Just an informal schema extraction or more in deep analysis? Do you count on possible linked schemas?

To summarise, I think that you offer a nice platform but for a paper (even for a poster/demo) there are too many unclear things and I don't see a clear advance in the state of the art.

**Anonymity:**

Yes, I would like my review to remain anonymous.

---

### Official Review · AnonReviewer3 · 2021-04-14
**Fail to see exactly the semantic and the interoperability dimensions of the approach**

**Rating:** 4
**Confidence:** 4

**Review:**

The authors propose a marketplace to semantically publish and consume datasets/datasources related to smart cities in order to enhance the usability and the interoperability of applications by enforcing FAIR principles. They also raise five challenges they faced (usual in semantic data management), but I fail to see how they have actually tackled them with the proposed architecture (I haven't been able to find further details in the provided webpages about the project itself).

While I find the project interesting, there may currently be too many questions arise about the current state of the solution:
- The maketplace is oriented to semantically annotate the datasources for publication and consumption. However, the data remains in the formats that the different stakeholders adopt, so the I of FAIR would be compromised. Even if I find a dataset about the same issue thanks to the annotations, the inner schema might differ and break my application. Which techniques does the project apply to ensure FAIR? There is an intermediate data format, but this is not enough to ensure interoperability.
- In order to publish the data sources, which models are you using? The variety of formats, access methods, schemas/ontologies followed by the actual data, ... might be a problem to find the appropriate data source.
- How does the marketplace infer the schema of the data when the providers publish their sources? It's stated that an initial semantic model is generated, but given that the source might be an FTP server, a data stream, ... how is the system doing this?
- When the providers publish their data, they can modify the conceptual model, adding and removing concepts and relationships on-demand. However, this might go against the shared dimension of ontologies: unilateral changes in a conceptualization can break the interoperability. How does the system deal with the alignment of collaboratively / concurrently derived schemas?


Typo:
page 1: this data => these data?

**Anonymity:**

Yes, I would like my review to remain anonymous.

---

### Official Review · AnonReviewer4 · 2021-04-14
**Marketplace for smart city semantic data, with an interesting feature**

**Rating:** 6
**Confidence:** 3

**Review:**

In this demo, the authors describe and propose a semantic data marketplace that aims at enabling a FAIR data sharing around the topic of smart cities. The marketplace allows cities, enterprises and citizens to share their data, and ensure that smart city applications follow the FAIR principles. The system extracts data schemas from supplied datasets and supports its evolution and customisation, therefore enabling more precise extraction and search. The marketplace currently contains 77 datasets from 3 cities.

The demo is well described and the paper accounts for what will be shown on the floor. The accompanying screenshot depicts the schema extraction in the specific example of data from a weather station. This feature has a lot of potential if proven to be precise and generalizes well to datasets of any sort.

On the other hand, important parts of the marketplace remain unclear. The method used for the semantic model extraction itself is not named, described or referenced. Similarly, the paper mentions that the semantic models are stored in a knowledge graph, but not so the data. How does this comply with the FAIR principles? This raises the bigger question of the paper: how does the marketplace ensure that data therein is indeed FAIR? This could be shown with e.g. FAIR metrics. If, as mentioned by the authors, publishing data as LOD does not make it automatically FAIR, the marketplace should have mechanisms in place to make it so, as it seems to be implied. Besides these, there might be an opportunity to improve the automatic extraction of semantic models, in the sense that these could be extracted using not just the data provided, but also schemas previously extracted from other marketplace datasets.


**Anonymity:**

Yes, I would like my review to remain anonymous.

---

### Official Review · AnonReviewer1 · 2021-04-16
**A data platform for semi-automated semantic annotation**

**Rating:** 5
**Confidence:** 4

**Review:**

This paper describes a data platform for semi-automated semantic annotation of datasets.

It shows a nice UI for managing and semantically annotating multiple data sets. The platform is capable proposing an initial semantic model for individual datasets based on their syntactic schema.

Besides the semantic data annotation capabilities, in its current state, I do not see any other added-value of this platform compared to traditional open data portals. It is unclear if the market place is leveraging semantic technologies to perform for example, concept searching and text auto-completion lookups to find relevant results. Also data views seem only possible over individual data sources, leaving the I in the FAIR principles unattended.

Given the scope of ESWC, I would suggest to the authors to clarify where and how their solution makes use of semantic technologies and what challenges had to be addressed.

**Anonymity:**

Yes, I would like my review to remain anonymous.

---

### Decision · Program_Chairs · 2021-04-19

Reject